# Obesity and Life History: The Hypothesis of Psychological Phenotypes

Amelia Rizzo [1,*] and Aldo Sitibondo [2]

1  Department of Clinical and Experimental Medicine, University of Messina, 98100 Messina, Italy
2  Infectious Diseases Unit, University Hospital "G. Martino", 98100 Messina, Italy; aldo.sitibondo@gmail.it
*  Correspondence: amrizzo@unime.it

**Abstract:** The aim of the present study is to postulate the existence of psychological phenotypes associated with obesity, based on individual history. While metabolic phenotypes have been acknowledged in the field of medicine, the same cannot be affirmed in the realm of psychology. A longstanding tradition in obesity research has sought to identify shared characteristics among individuals affected by obesity, including personality traits. However, research found no adequate empirical evidence to support the existence of a specific psychological and psychopathological profile among individuals with obesity. Recent efforts in the literature have attempted to correlate these findings and ascertain which metabolic phenotype correlates with a diminished quality of life. We propose a novel differentiation between two categories: (1) individuals who affected by obesity since childhood, and (2) individuals who developed obesity following a life event. Further investigations are imperative to amass experimental data that substantiate this hypothesis. Proactively identifying psychological phenotypes is presumed to impact therapeutic outcomes.

**Keywords:** obesity; life events; phenotypes





## 1. Introduction

Obesity is often defined as an excessive presence of adipose tissue in the human body, leading to a significant increase in health risks. The simplistic view of obesity as an energy imbalance, where energy intake exceeds energy expenditure, fails to acknowledge the complexity of this condition. The etiology of obesity involves intricate interactions among various genetic, environmental, and psychological factors [1].

Obesity is widely acknowledged as a risk factor for numerous diseases, including hypertension, dyslipidemia, cardiovascular disease, type 2 diabetes, arthritis, certain cancers, and depression. It is also associated with reduced life expectancy, diminished quality of life, and poorer overall mental health. However, the underlying causes of obesity remain poorly understood, even among healthcare professionals. Studies have shown that a considerable number of primary care practitioners believe that physical inactivity, overeating, and high-fat diets were the primary causes of obesity [2–4].

One of the earliest reports that "separated" the causes of obesity was the recognition by Babinski in France [5] and Fröhlich in Germany [6] at the turn of the 20th century that disease at the base of the brain—hypothalamic damage—could produce a loss of appetite control and obesity. This was followed 12 years later with a description by the American neurosurgeon, Harvey Cushing [7], that a tumor of the pituitary gland could also produce obesity. Subsequently, in 1953, Jean Mayer [8] proposed use of genetic, traumatic, and environmental factors as phenotypes for obesity, and Bray and York [9] proposed an anatomic and etiological classification for "obesities" (1979).

Nonetheless, the Foresight report identified more than 100 different biological, psychological, environmental, and social factors that potentially contribute to obesity [10]. These factors comprise genetic predisposition, neurochemical imbalances, emotional facets

such as indulgence in response to monotony or seeking solace through eating, along with intricate societal influences that might originate in one's early years (such as the notion of completing one's food) and endure across a lifetime, thereby rendering the act of making health-conscious decisions considerably arduous [10].

Irrespective of the underlying cause, once weight gain occurs, the body exhibits resistance to weight loss efforts. A normal physiological response to weight gain involves a decrease in metabolic rate and an increase in hunger, aiming to maintain the new weight equilibrium. Consequently, even when individuals successfully lose weight, most individuals will experience weight regain, necessitating lifelong management to sustain the new normal weight. Due to this chronic nature, obesity necessitates ongoing management akin to other long-term conditions, but is accompanied by an unprecedented degree of stigmatization [11–13].

## 2. The Myth of Obesity Psychological Patterns

The initial attempts to establish a connection between obesity and psychological aspects can be traced back to Hilde Bruch's original psychoanalytic conceptualizations in 1948 [14], and later expanded upon by relational-systemic and cognitive-behavioral perspectives. The objective of these authors was to develop psychogenetic hypotheses of obesity, theoretical models that could elucidate and predict the onset of obesity based on previous psychological, psychosocial, and dysfunctional family conditions [15].

However, a scientific understanding of the relationship between obesity and psychological factors has significantly evolved over the years, particularly in terms of empirical and methodological approaches. The early empirical studies, influenced by the psychogenetic theories of obesity, aimed to identify psychological, psychosocial, and psychopathological similarities among individuals with obesity by comparing them to those of normal weight. The primary goal of these studies was to explain the etiology of obesity by examining the causal contribution of psychological factors. Yet, these initial empirical investigations failed to demonstrate psychological homogeneity among obese individuals (refer to "The psychogenic theory of obesity" by Burchinal and Eppright [16], 1959). The authors concluded that there was no personality, psychopathology, or psychological profile that could account for the development of obesity [17].

Despite extensive research, there is insufficient valid and reliable empirical evidence supporting the existence of a specific psychological and psychopathological configuration among individuals with obesity [18]. Further, Lykouras and Michopoulos [19] argued that "there is no single personality type characteristic of the morbidly obese".

Concurrently, subsequent to etiological studies, some researchers shifted their focus to examining the psychological and psychopathological consequences of obesity rather than its causes. The notion gained traction that personality disorders or psychopathological disorders were more likely outcomes of obesity rather than causes [20]. However, cross-sectional comparisons between groups of individuals with obesity and those of normal weight yielded inconsistent results. Some studies indicated that individuals with obesity experienced greater psychological distress compared to normal weight individuals, while others found that obesity appeared to have a protective effect against psychological distress. The significant variability in results stemmed from differences in sample characteristics and methodological approaches [17]. The inconsistency of findings also reflected the heterogeneous nature of obesity's impact. Indeed, the effects of obesity vary among individuals, with some experiencing severe psychological, emotional, and psychosocial problems, while others exhibit milder distress or potentially no disturbance at all. Overall, initial studies collectively suggested that obesity itself is not directly associated with psychological distress but, rather, this association is mediated and moderated by other factors. Consequently, research on the psychological and psychosocial correlates of obesity transitioned into the second generation, focusing on identifying factors within the heterogeneous population of individuals with obesity that increase the risk of developing psychopathology [17].

The most compelling evidence comes from studies conducted on large representative samples, primarily in the United States, which suggest that women with obesity, but not men with obesity, exhibit higher levels of depression and suicidal ideation compared to their normal-weight counterparts [21,22]. Being female, as indicated by the aforementioned studies, can be considered a risk factor. Furthermore, severe obesity (BMI > 40) and comorbidity with binge eating disorder (BED) consistently correlate with an increased risk of psychopathology. Research has also documented a range of psychological and physical vulnerabilities associated with obesity, including low self-esteem and negative self-perceptions, disturbances in body image [23], sexual problems [24,25], eating disorders [26], reduced social interactions [20], and early trauma [27].

Despite this evidence, which suggests great individual differences and individual cases, much of the psychological literature works to identify common features. This is also demonstrated by the amount of systematic reviews of personality and obesity [28–30], which often find inconsistent associations between the obesity condition with broad personality domains, but links pertaining to only some facets of these domains.

More recently, [31] conducted a systematic review of personality variables among people affected by obesity. Authors argued that several studies found significant differences in personality traits between obese and non-obese individuals, particularly in neuroticism, conscientiousness, extraversion, and openness to experience.

Nevertheless, extraversion and openness to experience were found by the review as distinguishing factors for individuals with obesity, although the association was not consistently positive or negative. Some studies indicated a statistically significant correlation between body mass index and these two personality dimensions, but the direction of the correlation varied. The heterogeneity within the population of people affected by obesity suggests important differences that may influence the association with personality traits.

## 3. Obesity Metabolic Phenotypes

In the field of clinical and experimental medicine, the opposite trend seems to be observed. In fact, several studies have highlighted individual differences, leading to the identification of a clinical classification of different types of conditions among people with obesity. The aim has been to understand the biological variations among individuals with obesity in order to provide personalized treatment and improve the likelihood of achieving favorable outcomes. Some experts in nutrition have proposed classifying patients with obesity into distinct phenotypes [32]. Specifically, four obesity phenotypes have been described [33]:

1.  Normal-weight obese (NWO): This phenotype is characterized by a normal body mass index (BMI) but a high percentage of body fat, typical of obese individuals. NWO individuals exhibit elevated levels of vascular inflammation, significantly increasing their risk of cardiovascular disease. This phenotype is more prevalent in women, particularly among those over the age of 55 with a BMI below 25. NWO individuals have elevated blood pressure, fasting glucose, and poorer lipid profiles compared to healthy normal-weight individuals. They also tend to exhibit sarcopenia, particularly in the lower limbs. While they do not show metabolic syndrome, they experience high levels of oxidative stress and have a lower basal metabolic rate;

2.  Normal-weight metabolically obese (MONW): MONW individuals have a normal BMI and body weight, as well as a normal basal metabolic rate (BMR). However, they possess metabolic characteristics that predispose them to developing metabolic syndrome. This phenotype is characterized by features typically associated with obesity, such as high visceral fat, increased fat mass, low lean mass, reduced insulin sensitivity, elevated triglyceride levels, fatty liver disease, and chronic degenerative diseases. MONW is more prevalent in men and is commonly observed in women with polycystic ovary syndrome (PCOS). These individuals often exhibit smoking habits, hypertension, low physical activity levels, high levels of inflammatory markers

(C-reactive protein, tumor necrosis factor-alpha, IL-6), and low levels of high-density lipoprotein (HDL) and adiponectin;

3.  Metabolically healthy obese (MHO): The MHO phenotype represents individuals with clear obesity features but without any metabolic abnormalities. These individuals exhibit excess adipose tissue and a metabolic profile characterized by high insulin sensitivity, optimal lipid profiles, and absence of hypertension. They have a normal BMR and normal inflammatory markers. MHO is often observed in healthy post-menopausal women. When compared to healthy normal-weight individuals, MHO individuals show slightly higher levels of fasting glucose, triglycerides, C-reactive protein, low-density lipoprotein (LDL), systolic blood pressure, lower fiber intake, and reduced physical activity. Nevertheless, these individuals still face an increased risk of developing cardiovascular disease within 15 years of MHO diagnosis and metabolic syndrome within 10 years. Furthermore, subjects with MHO have been found to have hepatic steatosis (fatty liver) and elevated levels of high-sensitive C-reactive protein (hs-CRP);
4.  Metabolically unhealthy obese (MUO): The MUO phenotype is characterized by high BMI, a high percentage of body fat, elevated visceral adipose tissue, low BMR, a high incidence of metabolic syndrome, type 2 diabetes mellitus, atherosclerosis, and cardiovascular disease. MUO individuals exhibit high insulin resistance, abnormal lipid profiles, fatty liver disease, and elevated levels of inflammation.

More recently, Acosta et al. [34] proposed a Precision Medicine for Obesity at the Laboratory of the Mayo Clinic, based on different phenotypes. Authors measured various variables related to body composition, energy expenditure, eating behavior, affect, and physical activity in 450 participants affected by obesity. Based on these measurements, the participants were classified into different obesity phenotypes, which can be described as follows:

1.  Hungry Brain: Upon commencing their meals, these patients seamlessly progress to subsequent courses, experiencing a lack of satiety. Typically devoid of initial hunger, their consumption initiates an inability to curtail the act of eating;
2.  Hungry Gut: These individuals partake in a conventional meal; however, within a span of one to two hours, they begin to experience renewed sensations of hunger. Evidently, the gastrointestinal system fails to transmit the signal of satiety to the brain;
3.  Emotional Hunger: They engage in eating behavior with the intent of seeking rewards and eliciting positive sensations. Clinicians refer to this phenomenon as food addiction;
4.  Slow Burn: These patients exhibit a metabolism characterized by reduced efficiency in caloric expenditure.

The primary outcome assessed was weight loss after 12 months. The results revealed that phenotype-guided treatment led to significantly greater weight loss compared to non-phenotype-guided treatment, suggesting that targeting specific biological and behavioral phenotypes can enhance weight loss through pharmacological interventions.

A comprehensive assessment of body composition and genetic variations could also aid in selecting the most suitable surgical approach and predicting the therapeutic response for individuals affected by obesity. The integration of comprehensive data on obesity phenotypes is transforming the healthcare system, shifting the concept of medicine from a solely curative approach to a "proactive medicine that is predictive, preventive, personalized, and participatory" [33].

Abiri et al. [35] made an initial attempt to explore the association between various obesity phenotypes and common psychiatric symptoms, as well as health-related quality of life (HRQoL). The findings indicated that when obesity is accompanied by metabolic disturbances, the link with mental health issues and lower HRQoL becomes more pronounced.

## 4. Individual Differences in Obesity History

According to the guidelines provided by the National Institutes of Health (NIH), individuals eligible for bariatric surgery are those with a BMI greater than 40 kg/m$^2$ or those with a BMI greater than 35 kg/m$^2$ accompanied by comorbidities [36,37].

As part of the intervention process, a pre-surgical psychological assessment is conducted to identify any major ongoing psychiatric disorders, particularly eating disorders. However, as recommended by the guidelines of the Italian Society of Surgery of Obesity (SICOB), the evaluation also involves gathering the individual's medical history regarding the development of weight problems [38–40].

In clinical practice, individuals respond differently to the question of when their weight problems began. Two distinct groups can be identified:

1. The first group consists of individuals who have experienced obesity throughout their lives and have been in a constant cycle of dieting or weight loss and regain;
2. The second group comprises individuals who were initially within the normal weight range but started to gain weight "after a stressful life event".

Consequently, there are two possible types of individuals affected by obesity:

(a) Individuals who have been affected by obesity since childhood, with a history of weight control failures that likely began in childhood.

These individuals tend to have a relatively stable poor body image [41].

(b) Individuals who developed obesity following a traumatic life event.

These individuals have experienced being within the normal weight range and may have used food as a coping mechanism for stress [42].

Among individuals affected by obesity, the most commonly reported life events, according to the Paykel Life Event Scale [43], include maternal/paternal bereavement, sibling bereavement, abortions, loss of a child, pregnancy/postpartum period [44], marriage, job loss, divorce/separation, fractures/accidents/health problems requiring hospitalization, and relocation to another city.

Although many patients with obesity report that difficulties in weight control began after a traumatic life experience, to date, there is a lack of studies exploring the potential presence of psychological subtypes among individuals affected by obesity based on their weight history. Most studies have focused on comparing obese individuals with those of normal weight [45] or obese individuals with and without binge eating disorder (BED) [46,47]. These different subgroups may have distinct motivations for seeking intervention and may follow different paths of adherence and compliance following weight reduction interventions, such as sleeve gastrectomy. Given these considerations, it is important to investigate whether there are different psychological phenotypes based on the individual history of individuals affected by obesity.

## 5. Obesity, Life Stressors, and Emotional Abuse

Research indicates that experiencing emotional or behavioral abuse during childhood increases the risk of obesity in adulthood. A meta-analysis conducted by Hemmingsson, Johansson, and Reynisdottir [48] examined 23 studies with a total of 112,000 participants. The findings revealed that childhood physical abuse increased the risk of obesity by 28 percent, emotional abuse by 36 percent, sexual abuse by 31 percent, and unspecified abuse by 45 percent. Moreover, individuals who experienced severe abuse had a 50 percent risk of obesity compared to 13 percent for those who experienced moderate abuse.

Hemmingsson [49] further hypothesized that stressful life events leave lasting effects that can manifest as physical illnesses later in life. Negative events are believed to induce physiological stress, leading to negative thoughts, emotions, and increased inflammation in various body systems. This physiological response, along with compromised immune function and metabolism, is associated with obesity.

Recent reviews addressed the association between traumatic events and obesity development, from the perspective of adverse childhood experience. Schroeder et al. [50]

found that adverse childhood experiences (ACEs) increase the risk of developing obesity in childhood. The findings are consistent with prior reviews focused on childhood and adult obesity, suggesting that ACEs may interfere with psychosocial and neuroendocrine development, leading to obesity due to impaired self-regulation, appetite, psychopathology, and disruptions within family dynamics.

However, the relationship between ACEs and childhood obesity is complex and nuanced. Different studies showed conflicting findings for individual ACEs, and associations were sometimes found only for specific subgroups of individuals.

Wiss and Brewerton [51] indicated a range of potential pathways connecting ACEs to obesity, including biological, behavioral, health behavior-related, and psychosocial factors. The impact of ACEs on BMI appeared to be independent of social factors, supporting the biological embedding theory. Health behaviors and a chronic stress response were frequently suggested as mechanisms, while cognitive functioning and perceived discrimination received less attention. Psychological factors related to emotion dysregulation and dissociation have also been discussed in relation to trauma and eating pathology.

Several researchers have proposed that emotional eating may serve as a coping response to stressful events. Importantly, it is not the event itself that triggers the behavioral response but rather the negative interpretation and emotional reaction to the event. The concept of interoceptive awareness encompasses both a lack of clarity regarding experienced emotions and non-acceptance of these emotions [52].

Recent studies have highlighted a strong association between interoceptive awareness and body image. Individuals who can effectively attend to their body's internal cues tend to report higher levels of positive body image [53]. Trusting one's internal signals, such as heart rate or feelings of discomfort or hunger, is associated with a more positive perception of one's body and lower concerns about being overweight [54].

## 6. Surgical and Treatment Outcomes

It is crucial to emphasize the significant impact of emotional factors on the success of interventions and long-term weight maintenance. Surgical procedures alone are insufficient to address the underlying dysfunctional attitudes that contribute to the problematic relationship with food. The literature indicates that bariatric surgery can sometimes be ineffective in terms of achieving weight loss [55]. While patients typically experience a 50–60% reduction in weight two years post-surgery, approximately 20% of individuals fail to achieve significant weight loss. Psychological factors often contribute to this failure [56]. Hence, despite the benefits of bariatric surgery, psychological and behavioral factors are likely to play a crucial role in postoperative outcomes.

Simultaneously, considering the emotional consequences of obesity, a study conducted at the UCSD Medical Center in America [57] assessed how surgery can influence psychological and emotional factors, including eating behavior, mood, anxiety, and quality of life. The findings demonstrated that, following the intervention, individuals tended to exhibit reduced overeating in response to negative emotions, accompanied by a decrease in depressive and anxious symptoms. Quality of life also improved, although the study, being a pilot study, only evaluated outcomes up to 6 months postoperatively.

Hence, it is evident that the psychological component is intertwined with a complex set of physical, psychological, and behavioral variables. While common elements can be identified in the "syndrome" of obesity, a comprehensive understanding and treatment of individuals can only be achieved by considering their unique personal history and inner experiences. For example, people affected by obesity who do not differ in self-administered personality tests show different response in unconscious processes revealed by projective methods [39].

Hence, individuals with different histories may have different psychological and psychotherapeutic needs. Therapists and mental health professionals have at their disposal evidence-based treatments such as CBT-OB therapy [58]. Considering these observations, however, some reflections can be raised. With regard to the population of people who

have ever suffered from obesity, the results of a pilot study suggest that they might benefit from cognitive-behavioral therapies focusing on emotional regulation, such as Marsha Linhean's DBT, especially for "distressing tolerance" and "crisis survival" skills [59]. Regarding people with obesity starting from life events, who have different characteristics and specific needs, a program focused on trauma healing, such as EMDR, might be more indicated [60] as demonstrated by recent study on the effectiveness on EMDR in the treatment of obesity [61]. Awareness of individual differences can point to increasingly targeted pre- and post-surgical prevention, and may decrease prejudice and care disparity in psychological treatments.

## 7. Limitations and Conclusions

The ability to identify potential psychological phenotypes during the clinical pre-surgical psychological assessment provides clinicians with a valuable tool to determine the most appropriate intervention and support strategies, ultimately increasing post-operative success.

These observations in the clinical psychology field can contribute to reducing the stigmatization surrounding individuals with obesity, as the incidence of life events highlights that obesity is influenced by various factors beyond overeating or a sedentary lifestyle. A holistic approach, including psychological factors and individual history, is strongly recommended and should be adopted in the psychological treatment and prevention of obesity, with specific attention to individual development and the psychological and emotional aspects related to patients' personal histories.

Some limitations must be also pointed out. The concept of metabolically healthy obesity has certain limitations, primarily since a significant portion of individuals initially classified as "healthy" will eventually transition to the "unhealthy" phenotype over time. This is attributed to the chronic and relapsing nature of obesity as a disease process.

Furthermore, while categorizing individuals into these two psychological phenotypes can provide some insights, it is important to acknowledge the complexity of obesity and its multifactorial nature.

The relationship between obesity and life events is not always straightforward. There can be situations where patients who are already obese have experienced traumatic life events, contributing to their current weight status. Additionally, the impact of life events on obesity can vary among individuals, and other factors such as genetics, socioeconomic status, and environmental influences may also play significant roles.

Furthermore, it is crucial to recognize that psychological factors involved in obesity are not solely determined by a single traumatic event but can be influenced by a multitude of factors, including individual differences, coping mechanisms, and resilience. Therefore, a more nuanced approach is needed to understand the complex interplay between psychological factors, traumatic experiences, and obesity.

Future research should aim to explore the diverse pathways through which traumatic life events can affect obesity and consider a broader range of psychological phenotypes, taking into account the various complexities and individual differences within the obese population. This would provide a more comprehensive understanding of the psychological factors contributing to obesity and guide the development of tailored interventions for effective treatment and support.

**Author Contributions:** A.R. and A.S. have contributed substantially to the work reported, have written, revised and agreed to the submitted version of the manuscript. All authors have read and agreed to the published version of the manuscript.

**Funding:** This research received no external funding.

**Institutional Review Board Statement:** Not applicable.

**Acknowledgments:** I would like to express my heartfelt gratitude and deep appreciation to George A. Bray, Medicine Emeritus at the Pennington Biomedical Research Center, Louisiana State University. His invaluable contribution to this review has been truly exceptional, and I am immensely grateful for his insights and suggestions. Bray's vast knowledge and expertise in the field of medicine, and in particular obesity, have greatly enriched the content and quality of this work. His dedication to advancing scientific research and his unwavering commitment to excellence are truly inspiring. I wanted to take a moment to express my sincere gratitude to Rossella Alfa, for our discussions on clinical cases, shared experiences, patient treatment, and the study of these issues concerning individuals affected by obesity. Your insights and expertise as a psychologist and psychotherapist have been invaluable.

**Conflicts of Interest:** The authors declare no conflict of interest.

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
