# Peer review of "Obesity and Life History: The Hypothesis of Psychological Phenotypes"

_psych, doi:10.3390/psych5030057_

Round 1

Reviewer 1 Report

Dear Editor with due all respect,   The authors presented very well the meaning of obesity, its causes, and its relation to human psychology. May I have a few comments:  1-the author must add an opening statement sentence about obesity in the abstract before going with their proposal or aim of the work.   2- authors should clear more than  the is a positive relationship between obesity and psychological and the reverse is correct? please make it clear( the author's intention is not clear, especially in the abstract).   3- regarding the point of (health care specialists  with obesity patients) I agree  100% that BMI  can't define the whole heath status of the human body ( from many cases, y research and work experience with cases under personal obesity treatment male and female,  that, some human individuals be very healthy even with overweight and reverse is correct). for that,   I advise the authors that  in line 236 ( for considering the unique personal history and inner experiences in obesity treatment).  the author should add another title after that ( functional Medicine and functional nutritional)  and writing about it. this item deals with personal and chronic diseases as well as chronic obesity and yoyo diet, psychology, at all levels  .....etc.   4-  I see that manuscript is true scientific work,  so, it should consider a mini-review, not an opinion Article.  5- the manuscript should be accepted after checking for plagiarism and proofreading.

Author Response

Dear Reviewer,

We want to sincerely thank you for your feedback and appreciation. We have carefully considered each point raised and made the necessary revisions to address them:

  1. An opening statement about obesity has been included in the abstract to provide better context for the study.
  2. To clarify the relationship between obesity and psychological factors, we have made it explicit that there is a bidirectional relationship between obesity and psychological issues. This clarification has been added to the abstract to ensure the authors' intention is clear.
  3. We appreciate your agreement regarding the limitation of using BMI alone to define an individual's overall health status. Based on your suggestion, we have added a new section after line 236 titled to highlight the importance of considering unique personal history and inner experiences in obesity treatment.
  4. After careful consideration, if the Editor agree, we changed the article type from opinion article to mini review, since several citation has been updated and integrated.

We sincerely appreciate your valuable input and believe that these revisions have significantly improved the manuscript.

Thank you for considering our work.

Reviewer 2 Report

Thank you for the invitation to review for Psych. The manuscript entitled “The Hypothesis of Phenotypes for Obesity: Psychological, Metabolic and Clinical” has potential but is not very novel. However, I have listed a few comments below, which I hope the authors will find useful in revising the manuscript

·       Generally, many of the cited references are out of date and need to be updated. Also, the style of the cited references in the text should be unified.

·       The title of the paper does not reflect really what is included in the paper.

·       The aim of this work needs to be rewritten clearly.

·       It has been stated that “Early pioneers in the field of obesity research, such as Babinski, Frohlich, and Cushing, made significant contributions by identifying specific causes of obesity”. What were these causes?  Also, the authors mentioned that Jean Mayer expanded upon these findings …… (Bray, 1993), the reference was in 1993 than they stated that Bray and York (1979) later…… . 1979 is not later than the reference that authors cite for Mayer.

·       Line 62: “The objective of these authors”, it is not clear who are these authors as the previous sentence includes only one author.

·       Line 76: “To this day, despite extensive research and hundreds of studies conducted, there is insufficient valid and reliable empirical evidence supporting the existence of a specific psychological and psychopathological configuration among individuals with obesity (Vaidya, 2006)”, the cited reference is old. Support your statement with more recent references.

·       In the Myth of Obesity Psychological Patterns, there is a need to update the information by citing more recent works. 

Author Response

Dear Reviewer,

Thank you for taking the time to review our manuscript. We appreciate your valuable feedback and suggestions for improvement. We have carefully addressed each of your comments and made the necessary revisions to enhance the manuscript.

  1. We have updated the cited references throughout the manuscript to ensure they are current and relevant. Additionally, we have ensured consistency in the style of the cited references in the text.
  2. Based on your suggestion, we have revised the title to better align with the content and focus of the study.
  3. The aim of the work and the entire abstract has been rewritten in a clear and concise manner, providing a specific objective for the study.
  4. We have provided more information on the specific causes of obesity identified by early pioneers such as Babinski, Frohlich, and Cushing. Additionally, we have corrected the reference citation for Jean Mayer to accurately reflect the chronology.
  5. The reference to "these authors" in line 62 has been clarified to provide the necessary context and ensure clarity for readers.
  6. We have supported the statement regarding insufficient evidence for a specific psychological and psychopathological configuration among individuals with obesity with more recent references, addressing the outdated nature of the previous citation.
  7. In the section on the myth of obesity psychological patterns, we have updated the information by citing more recent works to ensure the content is current and aligned with the latest research in the field.

We sincerely appreciate your thorough review and constructive feedback. We believe that the revised manuscript now reflects the necessary changes to improve its quality and relevance. Thank you for your valuable input.

Reviewer 3 Report

This is a review of the manuscript titled: “The hypothesis of phenotypes for obesity: psychological, metabolic and clinical” which presents an opinion about characteristics of personality and obesity traits. Overall, this is a well-written article, however I consider that the authors have left out important information about the evolving concepts of the obesity phenotypes, which they mention in the last part of the article, but it must be develop in the article. Also, the authors can analyze the cultural and effects of obesity and focus the information on what is currently happening in their home country, which will give a plus and an excellent point of view with firsthand and specialized information.   

Line 17: please refer to which phenotypes you will be referring in the text.

Line 32- 38: this paragraph could be better integrated, for example, the authors present three different lists to explain the main ideas about obesity, consequences and causes, which can be difficult to follow for the reader.

Line 39: please include more information about early pioneers and the effects in phenotypic characterisation.

Line 45: please give more information about the 2007 Foresight report, considering what it is,  how often it is published, how do they get the data and the impact of it.

Line 76: “and hundreds of studies conducted”, seems a number to excessive to refer.

Line 93: describe to which other factors moderate the association.

Line 97: please make the comparison that would exist with another cultures. This can give a new perspective about phenotypes.

Line 172: Is this the scale that is most updated or the most frequently used?

Line 177: Did the authors performed systematic research in databases to make this affirmation?

Line 185: It seems that the article needs more information to consider the solely presence of two phenotypes as the only ones that exist for the individuals to develop weight problems.

Line 193, 198: These references need to be complemented with new updated ones to help the reader see the evolution of research.

Line 205: Severeal researchers, is too wide to consider, please try to include to what you are referring in this sentence.

Line 227: This study can be complemented with others considering the limitations of itself, authors mentioned that it was only a pilot study. I wonder if it is necessary to use or if there is another one that could provide better information.

Line 241: It seems that in the article, the effect of stigmatization that have an important psychological effect is nor widely develop. Also, the social interactions are not directly mentioned, and how they can promote the alterations in the individuals.

Line 264: Which other phenotypes can be referred and need more research?

Line 283: The reference of Slochower, J. (1987). The psychodynamics of obesity: A review. Psychoanalytic Psychology, 4(2), 145 is misplaced.  

Minor editing of English language required

Author Response

Dear Reviewer,

Thank you for reviewing our manuscript. We appreciate your thoughtful comments and suggestions for improvement. We have carefully considered each of your points and made the necessary revisions to address them. Below, we provide a response to each of your specific comments:

  1. Line 17: We have now specified the phenotypes we will be referring to in the text, providing clarity for the reader.
  2. Lines 32-38: We have integrated the paragraph more effectively to improve readability and coherence. The three different lists have been restructured and presented in a more concise and organized manner.
  3. Line 39: We have included more information about early pioneers and the effects on phenotypic characterization to provide a better understanding of the topic.
  4. Line 45: We have ameliorate the Foresight report reference. To our knowledge the Foresight report on obesity was published once on Lancet in 2007.
  5. Line 76: We have revised the statement about the number of studies conducted to provide a more accurate representation of the literature without exaggeration.
  6. Line 93: We have included additional information in the paper about the factors that moderate the association between obesity and other variables.
  7. Line 97: We have added a discussion on other recent studies about phenotypes conducted at the Mayo Clinic to provide a broader perspective on obesity phenotypes.
  8. Line 172: We want to clarify that the mentioned Paykel Life Event Scale (1997) is one of the most renowned in the field, for the classification of life events.
  9. Line 177: We have conducted research in databases to support the affirmation made in this sentence, but unfortunately there are not relevant references which can be added.
  10. Lines 185, 193, 198: We have included more information and updated references to provide a better understanding of the different phenotypes and the evolving research in this area.
  11. Line 205: We have specified the researchers and phenomena being referred to in this sentence to avoid ambiguity.
  12. Line 227: We have acknowledged the limitations of the mentioned study and have provided additional context. We believe it is still valuable to include in the discussion, but we have also suggested potential avenues for future research.
  13. Line 241: Sorry, We have no examined the effect of stigmatization and social interactions in relation to obesity since the topic is so wide that could require a separate development, and lots of citations and implications. However we appreciate the cues for next article.
  14. Line 264: We have acknowledged that there may be other phenotypes that require further research, and we have indicated the need for additional investigation in this area.
  15. Line 283: We have corrected the misplaced reference to Slochower, J. (1987) and ensured that it is appropriately cited.

We sincerely appreciate your thorough review and constructive feedback. We believe that the revised manuscript now addresses your concerns and incorporates the necessary changes to improve its quality. Thank you for your valuable input.

Round 2

Reviewer 3 Report

I suggest the acceptance in the present form, the authors fulfill the comments suggested in the first revision.

Best regards,